# The Design and Application of an Assistive Hip Joint Exoskeleton for Tower Climbing

**DOI:** 10.3390/s24237513

**Published:** 2024-11-25

**Authors:** Ming Li, Hong Yin, Zhan Yang, Hongwei Hu, Haoyuan Chen, Zhijun Fu, Xiao Yang, Zhao Guo

**Affiliations:** 1State Grid Hubei Electric Power Co., Ltd., Extra High Voltage Company, Wuhan 430050, China; lim260@hb.sgcc.com.cn (M.L.); 13707269857@163.com (H.Y.); 13071209048@163.com (Z.Y.); general712@163.com (H.H.); hychen@whu.edu.cn (H.C.); 2School of Power and Mechanical Engineering, Wuhan University, Wuhan 430072, China; 2019302080047@whu.edu.cn (Z.F.); 2019302080050@whu.edu.cn (X.Y.)

**Keywords:** tower climbing, hip joint exoskeleton, climbing motion, neural network

## Abstract

In order to ensure the safety of maintenance personnel during tower climbing and improve the efficiency of power maintenance work, this study designed an assistive hip joint exoskeleton robot and analyzed the kinematic data obtained from tower climbers during the climbing process. A neural-network-based assistive control algorithm for tower climbing was created, and a tower climbing experiment was conducted with volunteers. The surface electromyographic (sEMG) signals of four muscles, namely the biceps femoris (BF), gluteus maximus (GM), semimembranosus (SM), and semitendinosus (ST), were collected to evaluate the performance of the robot. The experimental results show that the exoskeleton robot could reduce the root mean square (RMS) values of the sEMG signals of the main force-generating muscles related to the hip joint. This suggests that the robot can effectively assist personnel in tower climbing operations.

## 1. Introduction

Currently, power maintenance operations are primarily conducted manually [1], and operators frequently need to climb towers to perform daily tasks [2,3]. Power pylons typically range from 50 to 60 m in height, with some of them even reaching up to 100 m. Tower climbing is physically demanding and poses a significant risk of physical exhaustion among maintenance personnel, which can lead to slipping and falling. In recent years, there have been numerous incidents resulting in personal injuries and fatalities due to falls from large heights. Therefore, there is an urgent need to introduce robotic technology to aid in such operations and allow maintenance personnel to conserve their physical strength [4,5].

While some specialized tower climbing robots have been developed in recent years [6], they still face several challenges, including slow climbing speeds, a limited ability to overcome obstacles, and poor environmental adaptability [7]. Therefore, the complete replacement of human labor using robots faces significant technical difficulties during high-level tower operations. The concept of the exoskeleton offers a better solution in the complex context of tower climbing [8,9]. An exoskeleton robot perceives the intention of the operator [10] and can provide an assistive force to the human body to reduce muscle torque [11]. This approach can reduce the physical strain on operators during tower climbing.

Exoskeletons can provide torque assistance for operators when climbing power towers, effectively reducing the risk of operational errors that may arise due to physical fatigue. The intelligent control of exoskeleton robots makes it possible to adjust the assistive force by estimating the state of the personnel; this can reduce climbing time and improve work efficiency, offering maintenance personnel a safe and comfortable tower climbing experience.

Thus far, exoskeleton robots have been used as assistive devices in various fields, such as healthcare [12,13,14,15], the military [16,17], and industry [18,19]. For example, Qian et al. developed a hip exoskeleton that can offer better actuation transparency and safety by using a nonlinear serial elastic actuator (nSEA) [20]. Juneil Park developed an assistive hip exoskeleton that can reduce physical exertion during human walking [21]. Divekar et al. developed an adaptive controller that provides assistance to users in lifting, lowering, and carrying tasks [22]. De Groof et al. designed a torque-controlled hip exoskeleton for patients with cerebral palsy for gait training [23]. The knee exoskeleton robot designed by Chun-Yi Kuo provides a recurrent neural network to estimate the knee joint torque for assistive control during stair climbing [24]. Although exoskeleton robots have been designed for stair climbing, tower climbing using exoskeletons is still a challenge as it is different from stair climbing. During power pylon climbing, electrical maintenance personnel often need to perform repair and safety tasks. The tower climbing motion imposes a greater demand in terms of the motion range and force of the human body.

In this study, we designed a hip exoskeleton robot specifically for tower climbing to assist electrical maintenance personnel. We designed the robot by collecting kinematic data from tower climbers. We built a dynamic model of the human leg during the swing phase of tower climbing for assistive force calculation. A deep-neural-network-based assistive control algorithm was developed to help operators to climb power pylons more easily. Experiments were carried out to evaluate the performance of this robot during high tower climbing.

We outline the structural and control design of the exoskeleton robot, as well as the experimental protocols, in Section 2. In Section 3, we present the experimental results, and, in Section 4, we summarize the main contributions of this work.

## 2. Materials and Methods

### 2.1. Kinematic Analysis of Tower Climbing

In tower climbing operations, the primary goal of the assistive exoskeleton robot is to provide torque to enable climbers to conserve their physical strength. It is essential to ensure comfortable operation when wearing the exoskeleton, as well as ensuring the safety of the climbing process. Therefore, a kinematic analysis of the lower-limb movements during the climbing process is necessary. This helps to ensure that there are no restrictions in the structure of the proposed exoskeleton robot. The tower climbing action is illustrated in Figure 1.

During the climbing process, the joint angle range of the operator is as shown in Table 1. Compared to normal walking, the hip angle of the operator in the sagittal plane can reach a maximum of 140°. Additionally, because the tower has a triangular cone shape, the movement during tower climbing differs from that observed in stair climbing. In the frontal plane, the operator will climb at a certain angle, and, at the same time, the hip joint will exhibit a degree of internal/external rotation.

Therefore, to aid tower climbing, the sagittal plane of the hip exoskeleton robot was developed as an active joint, while the internal/external rotation and internal retraction/extension planes were designated as passive joints for free operation.

### 2.2. Design of the Tower Climbing Assistive Exoskeleton Robot

#### 2.2.1. Structural Design

Based on the kinematic analysis of the tower climbing process, the designed lower-limb exoskeleton has three degrees of freedom (DOFs) for the unilateral leg. Among them, the flexion/extension of the hip joint are controlled by motors, while the internal/external adduction and internal/external rotation passively follow the self-adjustment of the tower climber. To ensure that the exoskeleton robot does not hinder the climber’s movement, this design incorporates a lightweight material. The main body parts are composed of carbon fiber, nylon, and lightweight materials, while the key support components are composed of aluminum alloy materials. This design ensures sufficient strength while minimizing the mass of the exoskeleton, resulting in a final prototype that weighs only 3 kg. The overall structural scheme of the prototype is shown in Figure 2.

#### 2.2.2. Hardware Design

As shown in Figure 3, the exoskeleton robot is powered by a 24 V polymer battery. The actuator of the hip exoskeleton utilizes GO-M8010 motors (Hangzhou Yushu Technology Co., Ltd., Hangzhou, China), ensuring a low rotational speed and high torque. This motor is equipped with an encoder to test the angle of the robot. In addition, we installed an inertial measurement unit (IMU) on the worker to collect the human joint angle and angular velocity, and we also installed sEMG sensors to gather myoelectric information in order to evaluate the effects of the exoskeleton.

### 2.3. Exoskeleton Robot: Climbing State Classification

According to the climbing process, we divided the climbing state into four phases. In the supporting phase, the leg is upright, stepping on the foot pegs to support the human body in standing. At this time, the exoskeleton robot does not need to provide assistance. In the swing phase, the thigh carries out a flexion motion while searching for the next foot peg location. During this phase, the exoskeleton robot assists the hip joint. In the holding phase (approximately 80°), the wearer conducts maintenance and repair tasks. In the lifting phase, the exoskeleton robot needs to provide downward assistance to achieve fast climbing. When the hip angle stretches to a certain degree, the thigh is nearly vertical, and the exoskeleton will work in zero-torque mode until it returns to the support phase. The climbing state classification is shown in Figure 4.

### 2.4. Dynamic Modeling of Human Leg During Tower Climbing

Since the mass of the exoskeleton’s mechanical component around the human thigh is very light (only 280 g), the initial part of the hip joint is very small compared to the human leg. Thus, we built a dynamic model of the human leg during tower climbing in the swing phase, aiming to obtain the torque of the unilateral human leg for the torque control of the exoskeleton hip joint [25,26]. First, we constructed the dynamics of the single lateral human leg during the swing phase.

The dynamic equation can be expressed as follows:(1)$$M\left(q\right)\ddot{q}+C\left(q,\dot{q}\right)\dot{q}+G\left(q\right)=\tau ,$$
where $M\left(q\right)$ represents the inertia matrix, $C\left(q,\dot{q}\right)$ represents the centrifugal and Coriolis loading, $G\left(q\right)$ represents the gravitational loading, and τ represents the solved human joint torque.

For the dynamic analysis, during the swing process, we can neglect the foot motion, and only the motion of the thigh and calf is considered. Thus, the unilateral human leg can be modeled as a two-link mechanism. The human leg model is shown in Figure 5.

Let $q_{1}$ denote the angle of motion of the hip joint, and let $q_{2}$ denote the angle of motion of the knee joint. The mass of the thigh is $m_{1}$ (8.5 kg), and the mass of the calf is $m_{2}$ (2.2 kg). The length of the thigh is $l_{1}$ (0.506 m), and the length of the calf is $l_{2}$ (0.405 m). The moment during thigh lifting is calculated based on the dynamics of the human joint. The dynamic parameters of the single lateral leg can be obtained as follows:(2)$$M(\theta )=\left[\begin{array}{cc} \frac{1}{3}m_{1}{l_{1}}^{2}+m_{2}{l_{2}}^{2}+\frac{1}{4}m_{1}{l_{1}}^{2}+m_{2}l_{1}l_{2}\cos\theta_{2} & -\frac{1}{4}m_{2}{l_{2}}^{2}-\frac{1}{2}m_{2}l_{1}l_{2}\cos\theta_{2}\\ -\frac{1}{4}m_{2}{l_{2}}^{2}-\frac{1}{2}m_{2}l_{1}l_{2}\cos\theta_{2} & \frac{1}{3}m_{2}{l_{2}}^{2} \end{array}\right],$$
(3)$$C(\theta ,\dot{\theta })=\left[\begin{array}{cc} -m_{1}l_{1}l_{2}{\dot{\theta }}_{1}\sin\theta_{1} & \frac{1}{2}m_{2}l_{1}l_{2}{\dot{\theta }}_{1}\sin\theta_{2}\\ \frac{1}{2}m_{1}l_{1}l_{2}{\dot{\theta }}_{2}\sin\theta_{1}+\frac{1}{2}m_{2}l_{1}l_{2}{\dot{\theta }}_{1}\sin\theta_{2} & -\frac{1}{2}m_{2}l_{1}l_{2}{\dot{\theta }}_{1}\sin\theta_{2} \end{array}\right],$$
(4)$$G(\theta )=\left[\begin{array}{c} -\frac{1}{2}m_{1}gl_{1}\sin\theta_{1}-m_{2}gl_{1}\sin\theta_{1}-\frac{1}{2}m_{2}gl_{2}\sin(\theta_{1}-\theta_{2})\;\\ \frac{1}{2}m_{2}gl_{2}\sin(\theta_{1}-\theta_{2}) \end{array}\right].$$

### 2.5. Controller Design for the Exoskeleton Robot

Inaccurate trajectories can lead to the misjudgment of the user’s movement intention, resulting in negative effects of the exoskeleton. In this work, we designed a controller based on convolutional neural networks and a finite state machine, which can plan the individualized gait of the operator and adjust the exoskeleton’s torque online.

The convolutional neural network model was used for phase prediction [27]. The architecture of the convolutional neural network model included a convolutional layer, a fully connected layer, a transposed convolution layer, and a softmax layer. The input data consisted of the angle and angular velocity of the hip joint, and the output represented the human body’s climbing state.

As shown in Figure 6, the first step of this control process was to carry out offline neural network training, which involved the continuous collection of the wearer’s hip angle and angular velocity, as well as the current stage of the tower climbing. These data were imported into the convolutional neural network for training, and the trained model was saved to the STM32 control hardware for online use.

For online control, the IMU first read the angle and angular velocity of the human and imported the offline-trained neural network model for phase judgment. The judged phases were then fed into the finite state machine for the state switching and control of the exoskeleton robot. During the swing phase, the human hip joint moment for robot control was calculated using the dynamic model. During the lifting phase, a constant torque was applied to achieve fast climbing.

### 2.6. Experiments on the Exoskeleton Robot

The foot pegs on the electrical tower base measured around 400 mm to 450 mm. We employed a combination of on-site and lab experiments. For the lab experiment, we used a ladder, with each flight of stairs being 21 cm in height. We stepped on two flights at a time to simulate the height between two piles. Ladder climbing and tower climbing experiments are shown in Figure 7.

The climbing experiments were divided into two groups: wearing or not wearing the exoskeleton robot. Each group of experiments was conducted five times. The climbing actions were not performed consecutively. A normal male with a height of 1.75 m and a weight of 80 kg was recruited. In the real field experiments, the same division was used, i.e., with and without the exoskeleton. The experiments followed a standard tower climbing procedure. The subject was an electrical tower maintenance worker.

Surface EMG signals were collected to evaluate the climbing effect [28,29]. The main muscles of the human hip are the gluteus maximus (GM), semimembranosus (SM), semitendinosus (ST), and biceps femoris (BF). Before applying the sEMG sensors to the subject, the application site was wiped with alcohol to reduce the EMG acquisition impedance. The sEMG signal is an electrical signal that measures muscle activity after placing electrodes on the surface of the skin, with the intensity being proportional to the level of muscle movement [30,31]. When the human hip joint carries out different movements, the strength of the corresponding EMG signal varies, allowing the sEMG signal to reflect the level of muscle activation.

We used an eight-channel EMG system (Changfeng Technology Co., Ltd., Beijing, China) with a sampling frequency of 1000 Hz. When the hip joint was in motion, the sEMG signals were contaminated with noise due to the thickness of the thigh fat layer, muscle fatigue, and external interference, necessitating data filtering.

Bandpass filters at 30 Hz and 500 Hz were applied to the sEMG signals, followed by full-wave rectification. The overall RMS and IEMG values of the data were calculated to compare the effects of the two groups. Meanwhile, to better illustrate the experimental effect, the RMS was chosen in this study to reflect the changes in the sEMG signals of the muscles during the experiment. The RMS effectively represents the amplitude of the signal, suppresses noise to a certain extent, and quickly responds to changes in the signal.

The formula for the calculation of the RMS is as follows:(5)$$RMS=\sqrt{\frac{\sum_{i=1}^{N-1}{\left|x_{\left(i\right)}\right|}^{2}}{N}}.$$

N is the number of samples in each window. In this study, we used samples with a length of 20 to compute the RMS each time, with the extraction window shifted by 10 values at a time. The RMS curve obtained still had some noise interference. A fourth-order low-pass filter with a frequency of 5 Hz was used.

## 3. Results

### 3.1. Simulation Experiments

One set of raw data from the myoelectricity experiment is shown in Figure 8. 

Simultaneously, full-wave rectification was performed on the original data, and a window with a length of 20 and a sliding step of 10 was selected to extract the RMS values. The extracted data were then subjected to fourth-order low-pass filtering to better demonstrate the experimental effect. The results of the experiment are shown in Figure 9. 

From Figure 9, it can be seen that wearing the exoskeleton robot to perform a single climbing maneuver reduces the RMS values of the four muscles, namely the BF, GM, SM, and ST. This in turn illustrates that there is a reduction in energy consumption for these muscles. Meanwhile, to address the potential error in a single experiment, we conducted five climbing experiments for each group and calculated the integral value (taking the same time length, IEMG), root mean square value (RMS), and maximum value (MAXABS) of the EMG signals from the five experiments, calculating their averages. The experimental data are shown in Table 2.

From the table, it can be concluded that, when assisted by the exoskeleton robot, the BF shows a 0.91% reduction in the IEMG, a 9.95% reduction in the RMS, and a 28.61% reduction in the MAXABS; the GM shows a 1.31% reduction in the IEMG, a 3.92% reduction in the RMS, and a 15.21% reduction in the MAXABS; the SM shows a 0.06% reduction in the IEMG, a 0.92% reduction in the RMS, and a 15.96% reduction in the MAXABS; and the ST shows a 4.2% reduction in the IEMG, an 8.61% reduction in the RMS, and a 1.27% reduction in the MAXABS. It is worth noting that the effect of the exoskeleton-assisted force on the SM is not significant. Overall, the RMS data indicate that the muscle was relatively relaxed and that the average level of activity was low when the exoskeleton was applied.

### 3.2. On-Site Experiments

As shown in Figure 10, the angle data from the power maintenance personnel indicate that, when the exoskeleton is used, the time taken for the climbing movement is shortened to a certain extent, especially during the assisting phase. Similarly, it can be observed that, when using the exoskeleton, the joint angle of the human body is larger than when not wearing the exoskeleton. This is due to the fact that, during the swing phase, the exoskeleton provides an assistive force, leading to an increase in the amplitude of the human body’s lifting movement.

Similarly, we analyzed the EMG signals, and the results are shown in Figure 11. The image itself has a slightly different shape compared to the results of the simulation experiment. The reason for this is twofold: it is partly due to the fact that the subjects in the two experiments were not the same, and the EMG signals of the same muscle when responding to the same action performed by different people can vary significantly; moreover, it is partly due to the inherent differences between the simulation experiment and the on-site field experiment. However, as far as the experimental effect is concerned, it can be concluded that wearing the exoskeleton robot to perform climbing maneuvers can reduce the RMS values of the four muscles: the BF, GM, SM, and ST.

## 4. Conclusions and Discussion

In this study, an exoskeleton robot for tower climbing was proposed, and a neural-network-based control algorithm was provided. Validation experiments on the algorithm were conducted. The results show that exoskeleton assistance could reduce the RMS values of the sEMG signals from the force-generating muscles related to the human hip joint. This is beneficial for tower climbing and could reduce the risk of accidents due to physical exhaustion among tower climbers. However, it is worth noting that the sEMG sensors were only used to detect four muscles, the BF, GM, SM, and ST. The effects on other non-hip muscles could also be considered, and the assistive robot could be validated by recruiting more subjects. In the future, we will develop other advanced controllers for the robot in order to reduce the energy consumption of workers during tower climbing.

## Figures and Tables

**Figure 1 sensors-24-07513-f001:**
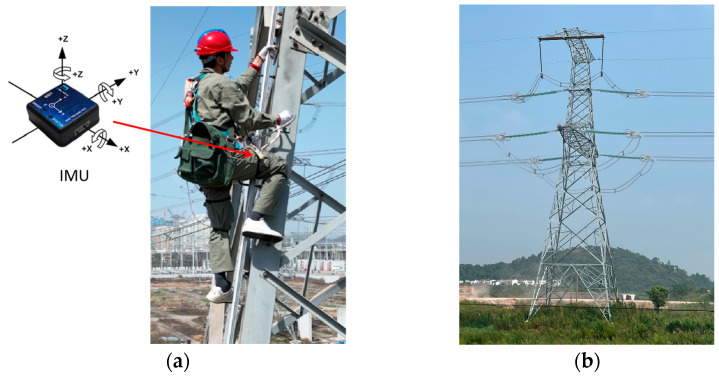
(**a**) Tower climbing motion and attitude data acquisition. (**b**) High-voltage power pylon.

**Figure 2 sensors-24-07513-f002:**
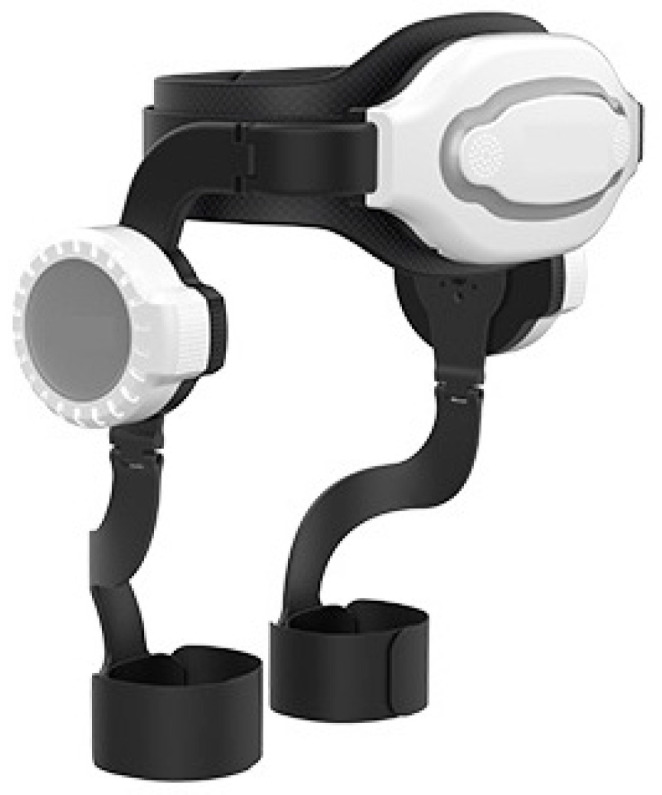
The hip joint exoskeleton robot.

**Figure 3 sensors-24-07513-f003:**
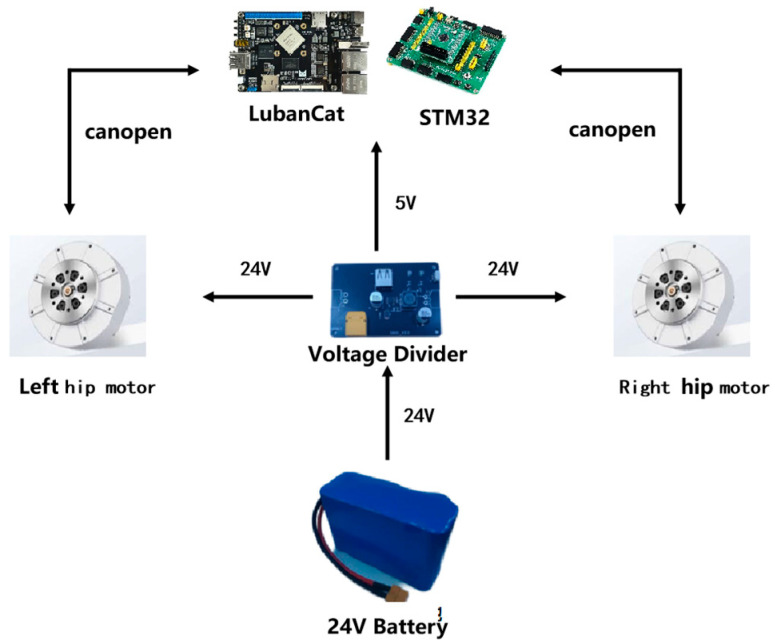
Exoskeleton hardware system.

**Figure 4 sensors-24-07513-f004:**
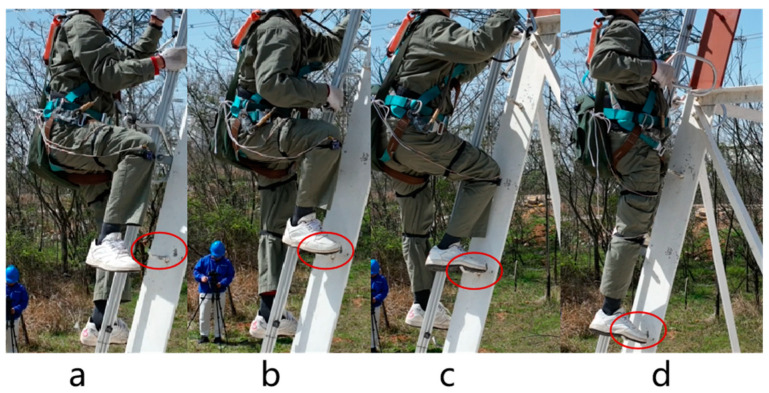
Climbing state classification: (**a**) swing phase; (**b**) holding phase; (**c**) lifting phase; (**d**) supporting phase.

**Figure 5 sensors-24-07513-f005:**
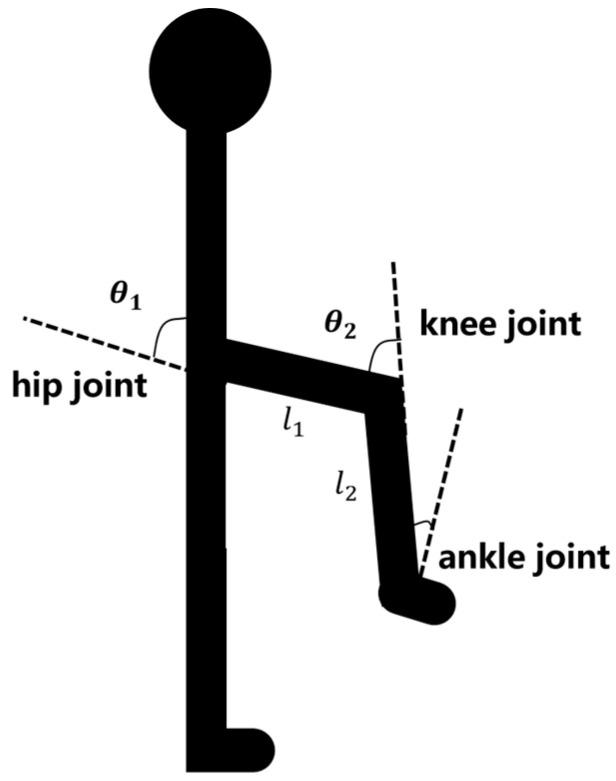
Schematic diagram of human leg model.

**Figure 6 sensors-24-07513-f006:**
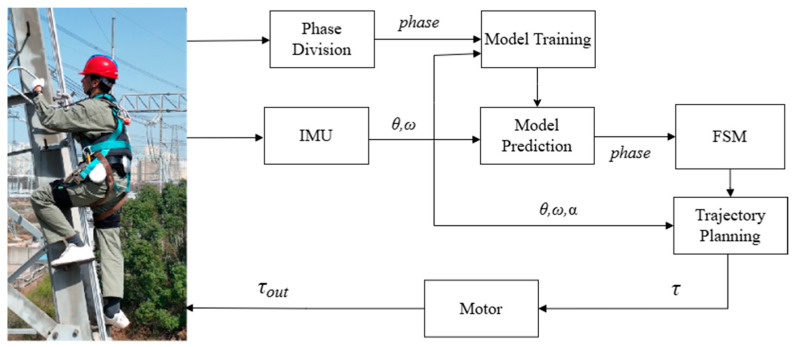
Tower climbing exoskeleton robot control diagram.

**Figure 7 sensors-24-07513-f007:**
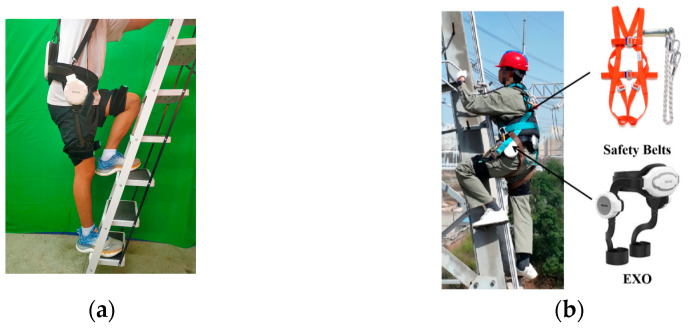
(**a**) Ladder climbing and (**b**) real tower climbing experiments.

**Figure 8 sensors-24-07513-f008:**
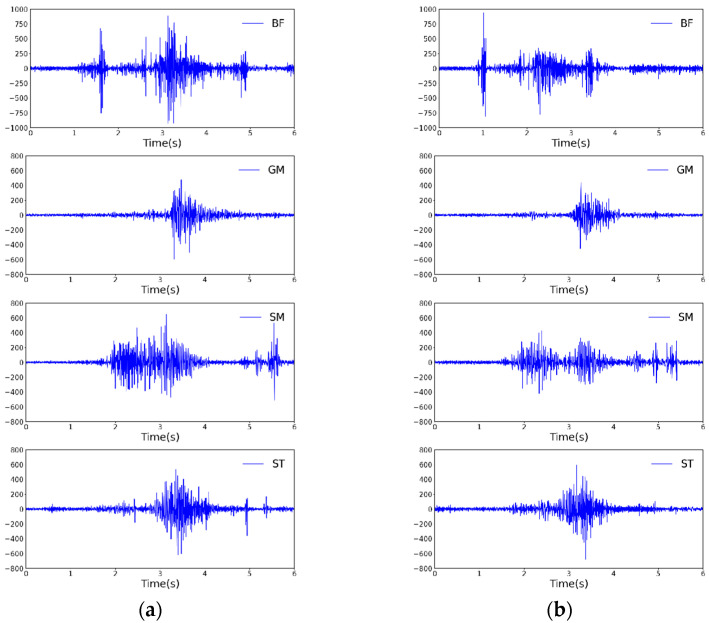
Raw experimental data.Here, (**a**) represents the EMG signals of the BF, GM, SM, and ST during a single climbing maneuver when the exoskeleton was not worn, while (**b**) represents the EMG signals of the BF, GM, SM, and ST during a single climbing maneuver when the exoskeleton was worn.

**Figure 9 sensors-24-07513-f009:**
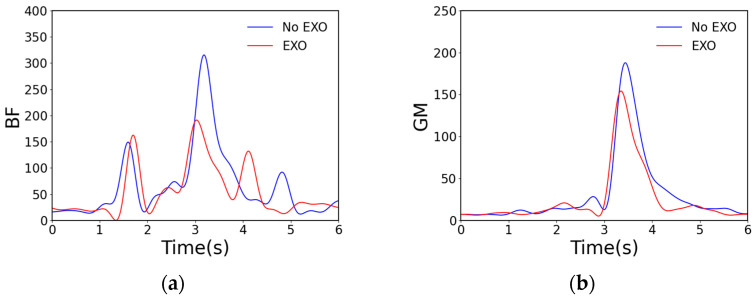
Comparison of RMS data for four muscles. Here, (**a**) represents the RMS comparison of the electromyograms of the BF with and without the exoskeleton; (**b**) represents the RMS comparison of the electromyograms of the GM with and without the exoskeleton; (**c**) represents the RMS comparison of the electromyograms of the SM with and without the exoskeleton; and (**d**) represents the RMS comparison of the electromyograms of the ST with and without the exoskeleton.

**Figure 10 sensors-24-07513-f010:**
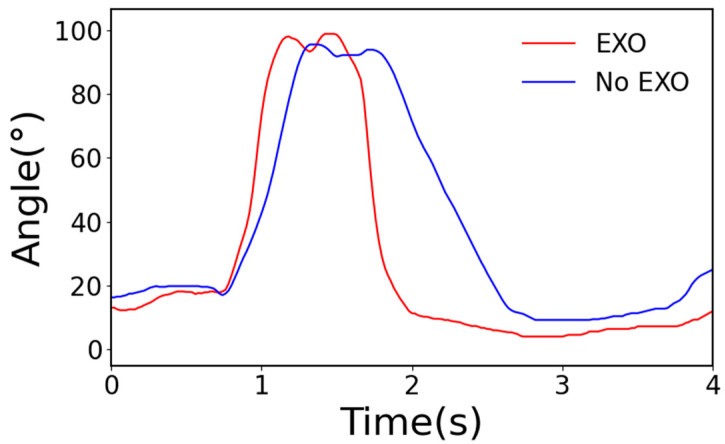
Hip angles during tower climbing.

**Figure 11 sensors-24-07513-f011:**
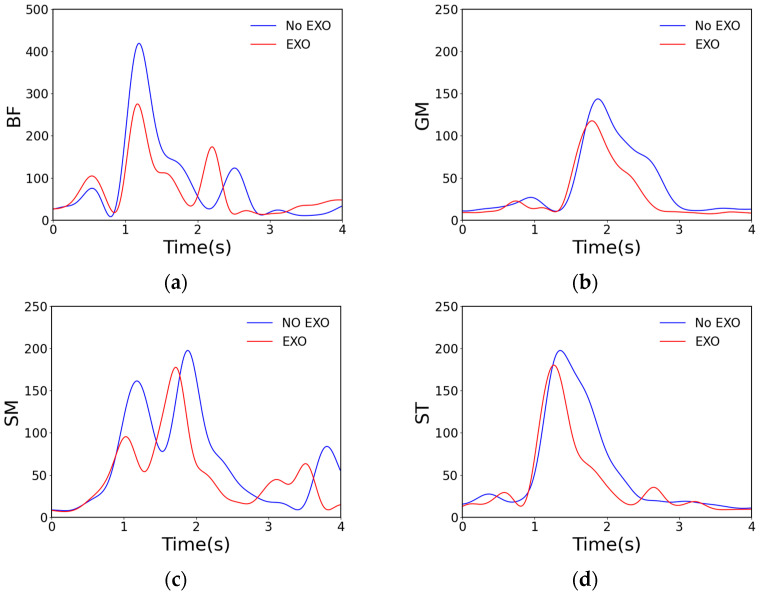
Comparison of RMS data for four muscles. Here, (**a**) represents the RMS comparison of the electromyograms of the BF with and without the exoskeleton; (**b**) represents the RMS comparison of the electromyograms of the GM with and without the exoskeleton; (**c**) represents the RMS comparison of the electromyograms of the SM with and without the exoskeleton; and (**d**) represents the RMS comparison of the electromyograms of the ST with and without the exoskeleton.

**Table 1 sensors-24-07513-t001:** The range of motion of the hip joint.

Process	Degree of Freedom Name	Range of Motion
Climbing process	Inward/outward	0~70°
	Internal/external rotation	0~20°
	Flexion/extension	−10~140°
Walking process	Inward/outward	−10~10°
	Internal/external rotation	−15~15°
	Flexion/extension	−30~40°

**Table 2 sensors-24-07513-t002:** Statistical sEMG signal data.

Muscle	Condition	IEMG	RMS	MAXABS
BF	NO EXO	293.39	95.59	1181.06
	EXO	290.695	86.08	843.17
GM	NO EXO	136.05	46.286	452.766
	EXO	134.274	44.47	383.886
SM	NO EXO	214.77	64.90	465.14
	EXO	214.64	64.30	404.86
ST	NO EXO	193.04	62.96	620.89
	EXO	184.85	57.54	612.98

## Data Availability

The data presented in this study are available on request from the corresponding author.

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
