# Peer review of "The Design and Application of an Assistive Hip Joint Exoskeleton for Tower Climbing"

_sensors, 2024, doi:10.3390/s24237513_

Round 1
Reviewer 1 Report
Comments and Suggestions for Authors
This paper designs a hip joint-assisted exoskeleton robot by analyzing the kinematic data and modeling dynamics of the climbing process of tower climbers. A neural network based tower climbing assisted control algorithm is also provided in the paper. It is verified by recruiting a volunteer that the exoskeleton robot can reduce the root mean square of the major hip-related sEMG signals of the force-generating muscles.
Overall, the design shows its innovation and engineering value. Meanwhile, the evaluation part has adequate experiment bases. However, there are still several drawbacks in the organization, illustration, and some minor mistakes in the passage. And the followings are details that may help to improve the paper.
Major issues:
1) In the introduction and literature review part, the authors have cited and analyzed lots of references. However, the problem the authors tried to embody is not shown clearly. Some analyses can be shortened. The novelty of the proposed tower-climbing assisted lower limb hip exoskeleton should be analyzed more clearly.
2) In the kinematics section, is it possible to represent the range of human hip joint movement angles during tower climbing and normal walking in more detail through a table?
3) Some definitions in the paper are not clearly explained. Can the division of stages during the ascending state process be described by a more specific division of angles to describe the four stages?
Minor issues:
1) Some sentences, such as the fourth sentence of paragraph 4 of the introduction, could be split into two sentences to improve readability.
2) More logical conjunctions can be used especially in the Introduction and Discussion parts to enhance the readability.
3) The legend of Figure 8 is not clear.
Author Response
Dear Editor-in-Chief, Technical Editor and Reviewers,
Thanks a lot for considering our paper for potential publication in sensors. We want to acknowledge all the constructive comments and suggestions, which helped us to significantly improve the quality of our paper and provided constructive suggestions for our future work. We have made revisions accordingly. Changes in the revised manuscript have been highlighted by yellow background in the revised manuscript.
Major issues:
- In the introduction and literature review part, the authors have cited and analyzed lots of references. However, the problemthe authors tried to embody is not shown clearly. Some analyses can be shortened. The novelty of the proposed tower-climbing assisted lower limb hip exoskeleton should be analyzed more clearly .
Response: We greatly appreciate your efforts of reviewing our submission. We apologize for not explaining the innovation of this paper clearly.We investigated a number of exoskeleton robots in industry, and it is worth noting that no exoskeleton robots have yet been used to assist in climbing electrical towers.In this study, we design an exoskeleton with a more suitable degree of freedom for climbing based on the kinematic data of the operation and maintenance personnel when they climb the tower, and we also design an exoskeleton control algorithm dedicated to the climbing tower booster.The innovation of this study is the design of an exoskeleton control strategy for assisted tower climbing.
- In the kinematics section, is it possible to represent the range of human hip joint movement angles during tower climbing and normal walking in more detail through a table?
Response: Thanks for your carefully detailed comments. In the new manuscript, we represent the range of human hip joint movement angles during tower climbing and normal walking in more detail through a table.
- Some definitions in the paper are not clearly explained.Can the division of stages during the ascending state process be described by a more specific division of angles to describe the four stages?
Response: Thanks for your constructive suggestion. In the new manuscript, we explained some definitions clearly,but we can’t describe ascending state process through division of angles because of the wide range of lifting angles between different people.
Minor issues:
- Some sentences, such as the fourth sentence of paragraph 4 of the introduction, could be split into two sentences to improve readability.
Response: Thank you for bringing up the mistakes. .In the new manuscript, we have modified it.
- More logical conjunctions can be used especially in the Introduction and Discussion parts to enhance the readability.
Response: Thank you for pointing out these omissions. In the new manuscript, we have modified these. And We have thoroughly review our manuscript to eliminate any grammatical and syntactical errors to improve the legibility of the manuscript. Revised sentences were highlighted in yellow.
- The legend of Figure 8 is not clear.
Response: Thank you for bringing up the mistakes. In the new manuscript, we have modified it.
Thank you again for your kind supports and we are looking forward to hearing from you shortly.

Reviewer 2 Report
Comments and Suggestions for Authors
The manuscript designed a hip joint-assisted exoskeleton robot for tower climbing and model dynamics of the climbing process of tower climbers. Moreover, a neural network based tower climbing assisted control algorithm was developed in this manuscript. As a result, the research has certain practical value in the field of power maintenance operations. However, the following points need to be clarified to improve the quality and the readability of the manuscript.
(1) It is well-known that exoskeleton robots have been widely used in many fields. The novelty of this paper with respect to the state of art is not clearly described. It is not easy to assess which is the main contribution of the work with respect to the present literature.
(2) The designed lower limb exoskeleton robot is a biped robot, which has 3 degrees of freedom for the unilateral leg. Single-leg dynamics of the robot is derived in this manuscript, but an important point within this work is that the gait planning and the interaction between the robot and human legs are not considered. And moreover, each leg of the lower limb exoskeleton robot is modeled as a two-link model, which is inconsistent with the fact that there are three degrees of freedom for the unilateral leg.
(3) The first two paragraphs of section 2.3 could be simplified because the Newton-Euler equations and Lagrangian equations are the basis of dynamic modeling for robots.
(4) There are some formulas in this manuscript, which are the well-known and basic results for two-link model of serial robots. Therefore, one suggestion is to simplify and put them in the appendix.
(5) At last, another suggestion is to revise English language and style to improve the quality and the readability of the manuscript. Unfortunately, there are many grammatical problems in the paper.

Comments on the Quality of English Language
Unfortunately, there are many grammatical problems in the paper. Thereofre, English language and style could be revised to improve the quality and the readability of the manuscript.
Author Response
1.It is well-known that exoskeleton robots have been widely used in many fields. The novelty of this paper with respect to the state of art is not clearly described. It is not easy to assess which is the main contribution of the work with respect to the present literature.
Response: We greatly appreciate your efforts of reviewing our submission. We apologize for not explaining the innovation of this paper clearly. We investigated a number of exoskeleton robots in industry, but it is worth noting that no exoskeleton robots have yet been used to assist in climbing electrical towers. In this study, we design an exoskeleton with a more suitable degree of freedom for climbing based on the kinematic data of the operation and maintenance personnel when they climb the tower, and we also design an exoskeleton control algorithm dedicated to the climbing tower booster. The contribution of this study is the design of an exoskeleton control strategy for assisted tower climbing. At the same time, we have also modified the introduction so that it better highlights our contributions
2.The designed lower limb exoskeleton robot is a biped robot, which has 3 degrees of freedom for the unilateral leg. Single-leg dynamics of the robot is derived in this manuscript, but an important point within this work is that the gait planning and the interaction between the robot and human legs are not considered. And moreover, each leg of the lower limb exoskeleton robot is modeled as a two-link model, which is inconsistent with the fact that there are three degrees of freedom for the unilateral leg.
Response: Thanks for your carefully detailed comments. We apologize for any misunderstanding caused by our negligence. In this part of the dynamics modeling, we want to derive not the exoskeleton dynamics, but the dynamics of the human unilateral leg. We want to solve for the moment of the human unilateral leg during flexion phase by using the Lagrangian method and input a percentage of this moment into the exoskeleton motors in order to reduce the energy consumption of the thigh lifting.
In this study, the tower climbing process was divided into four stages: flexion phase (a); footrest phase (b); booster phase (c); support phase (d). In the booster phase, a constant torque of 12 Nm is given to help the body achieve a fast climb. But in the flexion phase, the moment is solved by the kinetic model, and the exoskeleton outputs specific moments to assist the hip joint to perform the flexion action.This is due to the fact that the instantaneous application of constant force during the flexion phase can cause the body's center of gravity to become unstable. This is dangerous for tower climbing.
3.The first two paragraphs of section 2.3 could be simplified because the Newton-Euler equations and Lagrangian equations are the basis of dynamic modeling for robots.
Response: Thanks for pointing this out. In the new manuscript, we simplify the first two paragraphs of Section 2.3. We have also modified the entire dynamics section so that it is as free of misunderstandings as possible.
The dynamics model is analyzed and investigated mainly through the following two theoretical approaches: the Lagrangian mechanics and the fundamental theory of dynamics (including the Newton-Euler equations). In this paper, we choose to apply the Lagrangian method to solve the dynamics of single lateral leg flexion phase.
4.There are some formulas in this manuscript, which are the well-known and basic results for two-link model of serial robots. Therefore, one suggestion is to simplify and put them in the appendix.
Response: Thanks for your constructive suggestion. In the new manuscript, we have simplified this section.We removed some of the basic formulas and kept only the most essential ones.
5.At last, another suggestion is to revise English language and style to improve the quality and the readability of the manuscript. Unfortunately, there are many grammatical problems in the paper.
Response:Thanks for your carefully detailed comments. We apologize for any misunderstanding caused by our negligence. We have reviewed the manuscript thoroughly to correct any grammatical and syntactical errors.
Thank you again for your kind supports and we are looking forward to hearing from you shortly.

Reviewer 3 Report
Comments and Suggestions for Authors
Please see attached file.

Comments on the Quality of English Language
Although the reviewer may not be fully qualified to assess the quality of the English in this article, they recognize that the manuscript is generally well-written and logically structured. However, a detailed language review is recommended to enhance clarity and readability.
Author Response
1.To ensure greater clarity for the reader, the text in lines 31-32 could be written as: whilesome specialized tower-climbing robots have been developed in recent research [6], these robots still face several challenges, including slow climbing speeds, limited ability to overcome obstacles, poor adaptability of their end fixation mechanisms, and a tendency to remain at the experimental stage [7].
Response: We greatly appreciate your efforts of reviewing our submission. In the new manuscript, we have modified it in lines 30-33.
2.To comply with the journal's standards, these (and all) figures should be centered within the text.
Response: Thanks for your carefully detailed comments. In the new manuscript,We centered most of the figures and tables, but did not modify figures 1, 8 and 9 due to the layout requirements of the sensors journal.
3.To improve the overall presentation, it would be helpful to increase the spacing between the tables, 1 and 2, and the text.
Response: Thanks for your carefully detailed comments. In the new manuscript, We increase the spacing between the tables, 1 and 2, and the text to improve the overall presentation.
4.It is noted that the bibliographic references follow a mixed style, combining formats from journal articles and conference proceedings. For example, References [1] and [5]. To properly organize the references, please check the style required by the journal and follow its conventions accordingly.
Response: Thank you for bringing up the mistakes. In the new manuscript, we have standardized the format of the references, all in MDPI format.
Some remarks:
1.On Page 1, line 19, to ensure greater clarity for the reader, it may be helpful to add the meaning of the acronym RMS. The same to O&M in line 85; and IMU in line 125.
Response: Thank you for bringing up the mistakes. In the new manuscript, we add the meaning of the acronym RMS in line 16, the meaning of the acronym O&M in line 88, the meaning of the acronym IMU in line 88.
- On Page 1, line 19, replace“analysis;" with "analysis.".
- In line 55, replace "[27|et al." with "[27]."
- In line 55, replace "X. J." with "Jin and Guo"
- In line 57, replace "I. K." with "I. Kang et al "
- In line 275, replace "Fig.8." with "Fig.8”
- In line 283, replace "Fig.9." with "Fig. 9."
- In line 283, replace "figurel1" with "Fig. 11." Please try to follow a pattern throughout the
Response: Thank you for pointing out these omissions. In the new manuscript, we have modified these. And We have thoroughly review our manuscript to eliminate any grammatical and syntactical errors to improve the legibility of the manuscript. Revised sentences were highlighted in yellow.
Thank you again for your kind supports and we are looking forward to hearing from you shortly.

Round 2
Reviewer 2 Report
Comments and Suggestions for Authors
The manuscript has been revised, the corresponding relationship between the author's coverletter and the revision in the manuscript, however, is not given clearly in the presented coverletter. This leads to the consequence there is a great difficulty in judging whether the author's revision is appropriate for the reviewer. Meanwhile, the author's reply not only makes the reviewers see, but also makes the readers understand, and thus, the following points still need to be clarified to improve the quality and the readability of the revised manuscript.
(1) As the author stated, no exoskeleton robots have yet been used to assist in climbing electrical towers. Note that the first author of this manuscript comes from Ltd. Extra High Voltage Company, and therefore, one of my suggestions is that the practical necessity of using exoskeleton robots in climbing electrical towers assistance, from a practical point of view, could be strengthened in this manuscript.
(2) As far as the reviewer is concerned, my second question has not been completely dealt with in the revised manuscript. For example,
(i) single-leg dynamics of the robot is derived in this manuscript, but an important point within this work is that the interaction between the robot and human legs has not yet been considered so far;
(ii) note that the designed lower limb exoskeleton robot is a biped robot, which has 3 degrees of freedom for the unilateral leg. With regard to dynamic analysis of the robots, but each leg of the robot is modeled as a two-link model, which is inconsistent with the fact that there are three degrees of freedom for the unilateral leg. That is not sufficient and reasonable because of the coupling effect between the three joints. If only the thigh flexion process is considered, the research scope of the manuscript should be revised and limited, such as the title and the contribution because, the research in this manuscript is not enough to support the present title and the contribution.
(3) With regard to fig. 8 on page 8, the data should be further analyzed and explained.
(4) Further, English language and style still need to revised for the presented manuscript.
(5) And last but not least, the presented results, conclusions and discussion, as far as the reviewers are concerned, are not sufficient to support the research of lower limb hip joint assistive exoskeleton for tower climbing.

Comments on the Quality of English Language
English language and style still need to revised for the presented manuscript.
Author Response
1.As the author stated, no exoskeleton robots have yet been used to assist in climbing electrical towers. Note that the first author of this manuscript comes from Ltd. Extra High Voltage Company, and therefore, one of my suggestions is that the practical necessity of using exoskeleton robots in climbing electrical towers assistance, from a practical point of view, could be strengthened in this manuscript.
Response: We appreciate your efforts in reviewing our manuscript. In the revised manuscript, we have included the advantages of exoskeleton robot-assisted tower climbing in the introduction part. We also investigated research on exoskeleton robots for stair climbing and discussed the differences between stair climbing and tower climbing. Please see the revised manuscript.
“The use of exoskeletons as assistive devices can provide torque assistance for personnel climbing power towers, effectively reducing the risk of operational errors that may arise from physical fatigue. The intelligent control technology of exoskeleton robots can adjust the assistance based on the movement state of personnel while climbing the tower [12,13]. This adjustment reduces climbing time and improves work efficiency, offering maintenance personnel a safer, more flexible, and more comfortable tower climbing experience, which holds significant application value. ”
“The knee exoskeleton robot designed by Chun-Yi Kuo uses a recurrent neural network to estimate knee muscle torque for assistive control during the stair climbing process [24]. Although exoskeleton robots have been used in many fields, and some exoskeletons have been designed to assist with stair climbing, climbing a tower is not a rhythmic action like stair climbing. During the process of tower climbing, electrical maintenance personnel often need to perform repair and safety protection tasks. Therefore, the tower climbing motion imposes higher demands on the algorithms used in exoskeletons.”
2.As far as the reviewer is concerned, my second question has not been completely dealt with in the revised manuscript. For example.
(i)single-leg dynamics of the robot is derived in this manuscript, but an important point within this work is that the interaction between the robot and human legs has not yet been considered so far;
Response: Thank you for your detailed comments. We apologize for misunderstand caused by our negligence. In this part of section 2.3, we aim to obtain the needed human hip joint torque using the dynamic model of the single lateral leg. We do not modelling the exoskeleton dynamics for robot control. We would like to clarify the following points:
In this study, although all three degrees of freedom of the exoskeleton hip joint were designed, the degrees of freedom in the coronal and horizontal planes were only passive, with no actuation, and would be passively adjusted according to human movement. The exoskeleton can actively work in the hip sagittal plane. Additionally, since the mass of the exoskeleton hip joint is very light (only 280 g), the initial of the exoskeleton hip joint is small compared to human leg. Thus we aim to obtain the torque of the human unilateral leg during the swing phase for the exoskeleton hip joint torque control. Similar study on human dynamics has been carried out using soft exoskeleton for knee injury prevention During Squatting [1].
[1] Yu, S.; Huang, T.-H.; Wang, D.; Lynn, B.; Sayd, D.; Silivanov, V.; Park, Y. S.; Tian, Y.; Su, H. Design and Control of a High-Torque and Highly Backdrivable Hybrid Soft Exoskeleton for Knee Injury Prevention During Squatting. IEEE Robotics and Automation Letters 2019, 4, 4579-4586. https://doi.org/10.1109/LRA.2019.2931427.
[2] Scherb, David; Wartzack, Sandro; Miehling, Jörg. Modelling the interaction between wearable assistive devices and digital human models—A systematic review. Front. Bioeng. Biotechnol. 2023, 10, 1044275.https://doi.org/10.3389/fbioe.2022.1044275
(ii) note that the designed lower limb exoskeleton robot is a biped robot, which has 3 degrees of freedom for the unilateral leg. With regard to dynamic analysis of the robots, but each leg of the robot is modeled as a two-link model, which is inconsistent with the fact that there are three degrees of freedom for the unilateral leg. That is not sufficient and reasonable because of the coupling effect between the three joints. If only the thigh flexion process is considered, the research scope of the manuscript should be revised and limited, such as the title and the contribution because, the research in this manuscript is not enough to support the present title and the contribution.
Response: This is a simplification of the thigh and calf when the human body performs the leg lifting action. Since our exoskeleton can only assist the hip joint in the sagittal plane, we will only consider motion in the sagittal plane during swing phase, allowing us to simplify the thigh and calf into a two-link model.
In this study, the tower climbing process is divided into four stages: swing phase (a), foot rest phase (b), lifting phase (c), and support phase (d). During the lifting phase, a constant torque is applied to achieve fast climbing. During the swing phase, the human hip joint moment for robot control is calculated with the dynamic model.
3.With regard to fig. 8 on page 8, the data should be further analyzed and explained.
Response: Thank you for pointing this out. Fig. 8 shows the raw EMG signals acquired. Since EMG signals are highly susceptible to interference during acquisition, their signal processing is generally required. The surface EMG signal is an electrical signal that measures muscle activity by placing electrodes on the surface of the skin, with the intensity being proportional to the intensity of muscle movement. The processing of EMG signals is generally divided into two approaches: one is to remove interference through band-pass filtering, followed by full-wave rectification and the calculation of eigenvalues to reflect the intensity of EMG signals; the other is to represent the intensity of EMG signals graphically.
In this study, a band-pass filtering of the signal from 30 to 500 Hz is performed, followed by full-wave rectification. After that, the IEMG, RMS, and maximum values of the signal are calculated to compare the signal strength. Secondly, we chose to calculate the RMS values using samples of length 20, with a window offset of 10 values per extraction. However, some noise interference still exists in the root mean square curve obtained at this point. A fourth-order low-pass filter with a frequency of 5 Hz was used to process the data to enhance the clarity of the experiment.The processing of the data is described in Section 2.5.
References:
[1] Pradon, D.; Tong, L.; Chalitsios, C.; Roche, N. Development of Surface EMG for Gait Analysis and Rehabilitation of Hemiparetic Patients. Sensors 2024, 24, 5954. https://doi.org/10.3390/s24185954.
[2] Zhang, L.; Guo, Z.; Wang, C.; Yuan, Y.; Wu, X. Arm Motion Classification Based on sEMG and Angle Signal for A Lower Limb Exoskeleton Control System. In Proceedings of the 2019 2nd China Symposium on Cognitive Computing and Hybrid Intelligence (CCHI), Xi'an, China, 2019. https://doi.org/10.1109/CCHI.2019.8901935.
4.Further, English language and style still need to revised for the presented manuscript.
Response: Thanks for your carefully detailed comments. We have revised the manuscript with the help of a native English speaker. Changes in the revised manuscript have been highlighted by yellow background.
5.And last but not least, the presented results, conclusions and discussion, as far as the reviewers are concerned, are not sufficient to support the research of lower limb hip joint assistive exoskeleton for tower climbing.
Response: Thanks for this comments. We have conducted new experiments as shown in Section 3.2 of the revised manuscript. We evaluate the effectiveness of exoskeleton assistance using Surface EMG (sEMG) signals by placing electrodes on the surface of the skin. The root mean square (RMS) value of the detected EMG signal is small, means the muscle is less activated.
References:
[1] Yu, S.; Huang, T.-H.; Wang, D.; Lynn, B.; Sayd, D.; Silivanov, V.; Park, Y. S.; Tian, Y.; Su, H. Design and Control of a High-Torque and Highly Backdrivable Hybrid Soft Exoskeleton for Knee Injury Prevention During Squatting. IEEE Robotics and Automation Letters 2019, 4, 4579-4586. https://doi.org/10.1109/LRA.2019.2931427.
[2] Wu, P.; Chen, X.; He, Y.; Liu, Z. Unpowered Knee Exoskeleton during Stair Descent. In Proceedings of the 2021 7th International Conference on Mechatronics and Robotics Engineering (ICMRE), Budapest, Hungary, 2021. https://doi.org/10.1109/ICMRE51691.2021.9384832.
[3] Ma, X.; Long, X.; Yan, Z.; Wang, C.; Guo, Z.; Wu, X. Real-time Active Control of a Lower Limb Exoskeleton Based on sEMG. In Proceedings of the 2019 IEEE/ASME International Conference on Advanced Intelligent Mechatronics (AIM), Hong Kong, China, 2019. https://doi.org/10.1109/AIM.2019.8868817.
Thank you again for your kind supports and we are looking forward to hearing from you shortly.

Round 3
Reviewer 2 Report
Comments and Suggestions for Authors
The manuscript has been improved with respect to the previous version. Nevertheless, if only the thigh flexion process is considered, the research scope of the manuscript should be revised, such as the title and the contribution because, the research in this manuscript is not enough to support the present title and the contribution.
Comments on the Quality of English Language
English language and style still need to revised for the presented manuscript.
Author Response
Authors’ Response to the Review of Sensors-3272196
“The design and application of an assistive hip joint exoskeleton for tower climbing
”
Dear Reviewers,
Thanks a lot for considering our paper for potential publication in sensors. We want to acknowledge all the constructive comments and suggestions, which helped us to significantly improve the quality of our paper and provided constructive suggestions for our future work. We have made revisions accordingly. Changes in the revised manuscript have been highlighted by yellow background in the revised manuscript.
REVIEWERS' SUGGESTIONS:
- The manuscript has been improved with respect to the previous version. Nevertheless, if only the thigh flexion process is considered, the research scope of the manuscript should be revised, such as the title and the contribution because, the research in this manuscript is not enough to support the present title and the contribution.
Response: Thank you for your suggestions. We have modified the title and contributions of this revised manuscript according to reviewer’s comments.
“Title: The design and application of an assistive hip joint exoskeleton for tower climbing ”
“Contribution:In this study, we designed a hip exoskeleton robot specifically for tower climbing to assist electrical maintenance personnel. We designed the robot by collecting kinematic data from tower climbers. We built a dynamic model of the human leg during the swing phase of tower climbing for assistive force calculation. A deep neural network-based assistive control algorithm was developed to help operators to climb power pylons more easily. Experiments were carried out to evaluate the performance of this robot during high tower climbing..”
In this study, the tower climbing process is divided into four stages: swing phase (a), holding phase (b), lifting phase (c), and support phase (d). During the swing phase (thigh flexion process), the human hip joint moment for robot assistive control is calculated with the dynamic model. We aim to obtain the needed human hip joint torque. In the lifting phase, a constant torque is applied to achieve fast climbing, so we don't need to build the dynamics of this phase.
- English language and style still need to revised for the presented manuscript.
Response: Thanks for your carefully detailed comments. In the new manuscript, we will ask MDPI Author Services to provide professional English language editing.
Thank you again for your kind supports and we are looking forward to hearing from you shortly.
